# Essential Role of Astrocytes in Learning and Memory

**DOI:** 10.3390/ijms25031899

**Published:** 2024-02-05

**Authors:** Paula Escalada, Amaia Ezkurdia, María Javier Ramírez, Maite Solas

**Affiliations:** 1Department of Pharmaceutical Sciences, University of Navarra, 31008 Pamplona, Spain; pescalada@alumni.unav.es (P.E.); aezkurdia@alumni.unav.es (A.E.); mariaja@unav.es (M.J.R.); 2IdiSNA, Navarra Institute for Health Research, 31008 Pamplona, Spain

**Keywords:** astrocytes, calcium, synapsis, brain metabolism

## Abstract

One of the most biologically relevant functions of astrocytes within the CNS is the regulation of synaptic transmission, i.e., the physiological basis for information transmission between neurons. Changes in the strength of synaptic connections are indeed thought to be the cellular basis of learning and memory. Importantly, astrocytes have been demonstrated to tightly regulate these processes via the release of several gliotransmitters linked to astrocytic calcium activity as well as astrocyte–neuron metabolic coupling. Therefore, astrocytes seem to be integrators of and actors upon learning- and memory-relevant information. In this review, we focus on the role of astrocytes in learning and memory processes. We delineate the recognized inputs and outputs of astrocytes and explore the influence of manipulating astrocytes on behaviour across diverse learning paradigms. We conclude that astrocytes influence learning and memory in various manners. Appropriate astrocytic Ca^2+^ dynamics are being increasingly identified as central contributors to memory formation and retrieval. In addition, astrocytes regulate brain rhythms essential for cognition, and astrocyte–neuron metabolic cooperation is required for memory consolidation.

## 1. Introduction

The brain is the central organ responsible for various cognitive functions such as thinking, feeling, desiring, perceiving, learning, and memory. The exploration of learning and memory constitutes a significant area within neuroscience, with memory defined as a behavioral change resulting from experiences and learning characterized as the process of acquiring memory. Despite being extensively studied, the cellular mechanisms underlying these cognitive processes remain unclear. Interactions among local neuronal populations and connections across different brain regions are crucial for diverse types of learning [1]. Notably, neuronal action potentials occur on a short time scale, far from the time scale required for memory integration, while the timing characteristics of astrocytic communication and plasticity are more appropriate for broader time scales spanning hours, days, and months [2]. Although the focus in molecular and cellular research of learning and memory has predominantly been on neuronal mechanisms, latest investigations have delved into the roles of astrocytes in the field of learning and memory.

## 2. Role of Astrocytes in the Brain

Astrocytes, distinguished by their morphology resembling stars, are glial cells that establish close connections with neurons and blood vessels [3]. They wrap around numerous synapses and can form networks through gap junctions, creating well-organized anatomical domains with minimal overlap between neighbouring cells [3,4,5,6,7]. These distinctive morphological and phenotypic features make astrocytes strategically positioned to perceive their surroundings and adapt dynamically to environmental changes. Astrocytes play vital roles in various homeostatic and supportive functions (Figure 1), including maintaining glucose metabolism, monitoring glutamate and potassium levels, regulating cerebral blood flow, and secreting neurotrophic factors [8,9]. These established roles are essential for correct neuronal activity, as well as for the processes of learning and memory formation.

### 2.1. Astrocyte–Astrocyte Communication

Astrocytes form organized networks that are interconnected via gap junctions composed primarily of connexins (Cxs), especially Cx43 [6,10]. This allows a rapid exchange of ions and small molecules between adjacent astrocytes, playing an important role in the modulation of K^+^ and glutamate fluctuations, as well as in the propagation of Ca^2+^ waves, among others. Cxs can be hemichannels when they are not paired to adjacent ones, and they can regulate the extracellular release of gliotransmitters (ATP, glutamate, NAD, and D-serine). In addition, gap junctions can be open or closed depending on physiological or pathological conditions [10].

### 2.2. Interaction with Microglia and Oligodendrocytes

Microglia and astrocytes act together to coordinate neurogenesis, the elimination of extra synapses, and the regulation of inflammatory processes within the brain. In addition, astrocytes can release TGF-β, which induces the neuronal expression of C1q, an essential modulator of microglial pruning [11].

Oligodendrocytes are indispensable for maintaining myelin integrity. Astrocytes can contribute to this process by releasing oligodendrocyte growth factors that are transported via gap junctions [12,13], or by the direct modulation of myelination processes [14,15].

### 2.3. BBB Development and Maintenance and Neurovascular Coupling

Astrocytic endfeet are directly in contact with the blood–brain barrier (BBB) through endothelial cells and pericytes [16], as well as with neuronal synapses, rendering them essential for controlling blood flow changes depending on neuronal activity. Indeed, astrocytes express endothelial tight junction proteins and growth factors, which are necessary not only for BBB integrity, but also for neuronal development [16].

In the context of neurovascular modulation, these glial cells can also release gliotransmitters like glutamate, which leads to the neuronal release of nitric oxide or arachidonic acid derivatives that regulate blood flow depending on neuronal activity [17,18].

Astrocytic endfeet mainly express GLUT1 transporters and Kir4.1 and AQP4 proteins, stablishing a close contact with the brain capillaries. The GLUT1 transporter lets blood-borne glucose enter the astrocyte via a facilitated transport for neuronal energetic support. Kir4.1 is a potassium channel that maintains water and potassium levels, whereas AQP4 only regulates the water balance in the BBB [16,17].

### 2.4. Astrocyte–Neuron Metabolic Coupling

Astrocytes and neurons are genetically and metabolically different, although they cooperate to maintain brain energy demands [19]. Astrocytes exert higher glycolytic activity than neurons, whereas neurons exhibit higher oxidative rates.

According to astrocyte–neuron lactate shuttle (ANLS) theory, when astrocytes take glutamate, they increase ATP consumption via Na^+^/K^+^ ATPase and stimulate GLUT1 transporter activity [20,21]. At this point, astrocytes take blood-born glucose through their endfeet and convert it into pyruvate via glycolysis, which in the end will be transformed into lactate thanks to the LDH5 enzyme. This lactate is released into the extracellular space and taken up by neurons via monocarboxylate transporters (MCTs). Inside neurons, lactate is converted into pyruvate via LDH1 and used to obtain ATP through OXPHOS [22]. 

The glutamate–glutamine cycle is also an important way of reaching the metabolic requirements between neurons and astrocytes. Astrocytes take most of the glutamate released by neurons in the synaptic cleft, where it is converted into glutamine and taken up again by neurons to be used to synthesize new glutamate [23].

Other ways of astrocyte–neuron metabolic coupling are the exchange of NH4^+^ from neurons to astrocytes, which leads to enhanced astrocytic glycolysis [24], and the astrocytic transport of blood-borne glucose to neurons via the GLUT3 transporter [25,26].

### 2.5. Regulation of Synaptogenesis

The astrocytes of the grey matter or protoplasmic astrocytes possess specific endings called perisynaptic astrocytic processes (PAPs) that ensheathe thousands of synapses, what makes them key participants of synapse formation, function, and modulation [27].

There are diverse astrocytic components involved in synaptogenesis. The first one discovered was cholesterol. Neurons are able to synthesize cholesterol “de novo”, but they need an extra amount to really form functional and mature synapses. This extra cholesterol is supplied by astrocytes [28,29]. Interestingly, not only cholesterol helps synaptic maturation, but also cell adhesion proteins and growth factors (i.e., SREBP-1, TSP, BDNF, TGF-β) that are highly expressed in some astrocytes that regulate lipid metabolism [30,31]. More in detail, these molecules are synthesized in PAPs, near the synapses where they will be released [32].

### 2.6. Regulation of Synaptic Plasticity and Activity

Over the last twenty years, previously unforeseen roles of astrocytes have emerged, demonstrating their direct impact on neuronal activity. These roles encompass the regulation of synaptic transmission and formation, as well as neuromodulation and plasticity [33,34,35,36]. These functions are regarded as the foundational elements for information processing and the establishment of memory. The subsequent sections explore these newfound effects and contemplate their potential contribution to memory processes.

## 3. Bilateral Neuron–Astrocyte Interaction as a Potential Mechanism for Memory Modulation

Memory consolidation refers to the transformation over time of recently acquired information and their neurobiological underpinnings into stable memories. This transformation entails molecular alterations at the synaptic level leading to the reorganization of neuronal networks [37], believing that these changes in synaptic connectivity could be crucial for the acquisition and stabilization of new memories [38]. 

Some studies have shown the indispensable contribution of astrocytes to neurotransmission [39]. Their role in dynamically integrating and modulating synaptic information [40] and shaping plasticity [41] has become increasingly apparent. There are also increasing studies showing that astrocytes have an impact on behaviour [42].

### 3.1. Regulation of Synaptic Formation

Mounting evidence suggests the significance of neuron–glia communication in synaptic function, although the underlying mechanisms remain incompletely understood. In vitro studies carried out on purified neurons and astrocytes helped to understand how astrocytes take part in synapse formation. Astrocytes release different soluble molecules, such as cholesterol [43], and this release of substances increases the number of mature functional synapses and demonstrates that these cells are required for synaptic maintenance in vitro [28,29]. When the interaction between astrocytes and neurons is disrupted, it results in compromised synaptogenesis, both in vitro and in vivo [44]. This highlights the indispensable role of astrocytes in the actual formation of synapses. 

Additionally, the regulatory influence of astrocytes on synaptic development is not limited to the early stages of brain development. Evidence indicates that they also regulate the process of synapse elimination and influence the morphology of dendritic spines in the adult brain [45,46]. 

When considered collectively, these findings indicate the significant involvement of astrocytes in remodelling synaptic connections within the brain. The current research underscores the crucial role of synaptic structural plasticity in the process of memory formation [47]. These investigations reveal that the formation and elimination of synaptic connections occur within specific cortical neuron subpopulations during a variety of sensorimotor learning tasks [48,49]. Consequently, the regulatory impact of astrocytes on synapse formation and elimination represents a potential mechanism underlying the modulation of learning and memory.

### 3.2. Regulation of Synaptic Modulation

The astrocyte–neuron connection is not static but rather plastic, astrocytic processes show spontaneous morphological changes, and these dynamic changes in the astrocytic coverage of synapses can modulate synaptic transmission and contribute to brain signalling [50]. For example, synaptic transmission is influenced by the surface molecules expressed in the PAPs, such as the excitatory amino acid transporters 1 and 2 for glutamate transport [51].

These findings have contributed to the development of the “Tripartite Synapse” concept, suggesting that astrocytes are not just supportive structures around synapses. However, they actively communicate with both pre- and postsynaptic neurons, altering synaptic activity [52]. Astrocytes interact with synaptic neuronal components, responding to synaptic activity and subsequently modulating synaptic transmission. Evidence indicates that astrocytes play a crucial role in integrating and processing synaptic information, as well as in regulating synaptic transmission and plasticity. As active participants in synaptic function, astrocytes are cellular components engaged in the processing, transfer, and storage of information within the nervous system [40].

Astrocytes lack electrical excitability and do not generate action potentials. Nonetheless, astrocytic excitability is apparent through elevated cytosolic Ca^2+^ levels. Consequently, astrocytes can release gliotransmitters like glutamate, adenosine triphosphate (ATP), and D-serine in a Ca^2+^-dependent manner [36]. These gliotransmitters have the capability to bind receptors located on either the presynaptic or postsynaptic sites, modulating neuronal excitability or regulating synaptic transmission [53,54,55]. 

The release of astrocytic glutamate regulates neuronal Ca^2+^ levels and ultimately has a decisive impact on synaptic transmission. Currently, it is established that astrocytes detect synaptic activity by activating metabotropic or ionotropic receptors, responding to these signals with increases in intracellular Ca^2+^ levels [56]. These findings suggest that astrocytes could detect and potentially integrate synaptic signals, thereby indicating their potential involvement in information processing within the brain and suggesting a reciprocal communication between both cell types [53,57]

A single gliotransmitter can target various receptors, yielding multiple effects contingent upon the circuit type, targeted neurons, location of neuronal receptors (pre- or postsynaptic), and the subtype of the activated receptor [36]. For example, in the hippocampal dentate gyrus, astrocytic glutamate temporarily potentiates excitatory transmission by engaging presynaptic NMDA receptors [58], but also, the release of glutamate from astrocytes can stimulate various classes of glutamate receptors and selectively adjust inhibitory synaptic transmission [59]. 

The environment in which a synapse is embedded also determines the function and efficacy of the synaptic transmission. Thus, homeostatic and supportive functions such as lactate transport from astrocytes to neurons and potassium clearance are also important to synaptic activity [60,61].

Additionally, recent research also indicates that astrocytes might contribute to the development of synaptic networks by overseeing spine pruning [46]. Astrocytes have the ability to quickly extend and withdraw their processes to interact with and detach from dendritic spines [62]. The movement of astrocytic processes is controlled by the glutamate released at synapses, which activates metabotropic glutamate receptors on astrocytes and generates astrocytic Ca^2+^ transients. This process of movement and coverage of dendritic spines is linked to the stability of the spines. Hence, PAPs possess the machinery to both sense neuronal activity and modify their actin filaments in an activity-dependent manner, potentially to regulate the stability of new spines [63,64]. 

Thus, this capacity of sensing synaptic activity and releasing gliotransmitters allows astrocytic networks to integrate neural activity and influence action potential firing in neuronal networks, thereby influencing their spatiotemporal dynamics [65].

### 3.3. Neuromodulation

Astrocytes contribute to neuromodulation by expressing different types of receptors such as cholinergic, adrenergic, dopaminergic, and serotonergic receptors [56]. For example, the direct application of acetylcholine and the stimulation of cholinergic inputs in hippocampal slices result in increased intracellular Ca^2+^ levels in astrocytes by activating muscarinic cholinergic receptors [66]. 

Cholinergic activity in vivo induced by the electrical stimulation or sensory stimulation of the septal nucleus leads to an elevation in Ca^2+^ levels within hippocampal astrocytes. Cholinergic-induced long-term potentiation (LTP) relies on the increase in Ca^2+^ levels in astrocytes, which, in turn, trigger the release of glutamate by astrocytes, subsequently activating mGluRs. The cholinergic-induced LTP emerges from the temporal alignment between postsynaptic activity and the astrocytic Ca^2+^ signal, triggered by cholinergic activity. Thus, this increase in Ca^2+^ is essential for the cholinergic-induced synaptic plasticity [67].

### 3.4. Adult Neurogenesis

The phenomenon of adult neurogenesis, which involves the generation of new neurons in the adult brain, takes place within two primary regions: the dentate gyrus of the hippocampus and the olfactory bulb [68]. It has been shown that adult neurogenesis contributes to learning and memory [69], anxiety and stress regulation, and social behaviour [70]. There is evidence of disturbed neurogenesis in the hippocampus of patients with neurodegenerative diseases, such as Alzheimer’s disease [71].

Astrocytes release molecules, such as glutamate, ATP, and D-serine, which regulate various stages of adult neurogenesis [72]. These stages involve the proliferation of neural stem cells, their subsequent differentiation into neurons, the formation and integration of synapses, as well as the promotion of neuron survival [73,74,75]. 

In addition, astrocytes mediate the cholinergic regulation of adult hippocampal neurogenesis and memory via M1 muscarinic receptors. The deletion of CHRM1 in astrocytes led to defects in neuronal stem cell survival, neuronal differentiation, maturation and integration of newly formed neurons in the dentate gyrus, as well as impaired contextual fear memory [76].

Numerous studies have found a correlation between impaired neurogenesis, reduced cognitive performance, and irregularities in astrocytic function. Additionally, these studies have shown that treatments to improve cognitive function are often accompanied by an increase in neurogenesis and restoration of normal astrocytic morphology [77,78].

## 4. Implication of Astrocytes on Learning and Memory Processes

The modulation of synapses is widely recognized as a crucial mechanism in the processes of learning and memory. As highlighted in the preceding section, the role of astrocytes in these processes has long been theorized due to their ability to interact with synapses. Only in the past decade has substantial investigation into the involvement of astrocytes in learning and memory begun. Advances in molecular genetics have been pivotal in facilitating targeted interventions in awake, behaving experimental animals. These interventions encompass the use of cell-type-specific recombinase driver mice, an inducible expression system, specialized promoters designed for cell-type specificity in recombinant viral vectors, along with genetically encoded functional indicators and actuators, among other techniques [79,80].

In this context, we examine research investigating the roles of astrocytes in memory and learning, classifying them into (1) working and spatial memory (Table 1), (2) recognition memory (Table 2), and (3) contextual memory (Table 3).

### 4.1. Working and Spatial Memory

Astrocytes contribute to working memory by temporarily storing and manipulating information over brief intervals, usually lasting seconds. Considerable attention has been devoted to exploring the G-protein-coupled receptors (GPCRs) present in astrocytes. A fundamental mechanism for triggering heightened calcium (Ca^2+^) levels involves the activation of Gq-GPCRs. The elevation of astrocytic Ca^2+^ responses induced by Gq is associated with the activation of the phospholipase C (PLC)/inositol 1,4,5-trisphosphate (IP3) pathway as follows: upon GPCR activation, PLC cleaves the membrane lipid phosphatidylinositol 4,5-bisphosphate, producing diacylglycerol (DAG) and IP3. This leads to the activation of IP3 receptors (IP3R) and the subsequent release of Ca^2+^ from the endoplasmic reticulum (ER) [53]. The adjustment of Ca^2+^ levels has been shown to either enhance or diminish working memory capabilities.

Gq signalling pathway activation in hippocampal astrocytes through chemogenetic and optogenetic approaches has been demonstrated to enhance long-term working memory [81,82,83]. While these studies underscore the vital role of astrocytic activation in controlling memory performance, they do not explicitly illustrate astrocyte activation during short-term memory. In contrast, evidence for such activation has been provided by studies that suppress specific signalling pathways in astrocytes. For example, knockout mice lacking S100β (a Ca^2+^-binding protein) exhibit improved short-term spatial memory and heightened hippocampal LTP [84].

In contrast, the inhibition of astrocytic Gq-GPCRs leads to decreased memory performance in the Y-maze test, indicating defects in spatial working memory [85]. Correspondingly, transgenic mice lacking IP_3_R2 exhibit disrupted GPCR-mediated Ca^2+^ signalling and alterations in the Y-maze [86]. Additionally, mice with a genetic astrocytic deletion of Gq-coupled GABA_B_ receptors display impaired performance on the T-maze and reduced gamma oscillatory activity [81], further supporting the notion that Gq-GPCRs in the astrocyte play a key role in working memory.

These studies collectively suggest that the stimulation of Ca^2+^ through the activation of Gq-GPCRs supports working and spatial memory, whereas the inhibition shows the inverse effect.

In addition to manipulating Ca^2+^ levels, other mechanisms may also play a role in the modulation of astrocytic memory. For example, the removal of Gs-coupled adenosine A_2A_ receptors on astrocytes has been shown to impair spatial working memory and disrupt glutamate homeostasis [87]. Interestingly, another study reported contrasting findings, indicating that the elimination of these receptors in astrocytes leads to an improvement in short-term spatial and long-term contextual memory [88]. This suggests a dual function of astrocytic A_2A_ receptors, allowing for the precise adjustment of both short working memory and long contextual memory. Previous studies have demonstrated that A_2A_ receptors influence behaviour in different ways depending on the cell compartment and/or brain area they are expressed [89,90], probably linked to the diverse roles of adenosine from various sources as a regulator and modulator of synapses and brain function [91]. Future investigations should delve into the bidirectional regulation of working memory and long-term memory by astrocytic A_2A_ receptors.

Astrocytes play a crucial role in providing metabolic support for working memory. The ANLS model proposed by Pellerin and Magistretti postulates that astrocytic glucose glycolysis and neuronal oxidative phosphorylation cooperate in memory formation through lactate transport [92]. Indeed, the pharmacological inhibition of glycogenolysis has been shown to impair working memory, while the exogenous lactate administration is capable of rescuing memory deficiencies [93].

**Table 1 ijms-25-01899-t001:** Astrocyte involvement in spatial and working memory.

Proposed Astrocytic Mechanism	Outcome of the Astrocytic Manipulation	Test	Reference
Melanopsin-induced astrocytic optogenetic activation	Improved working and spatial memory	T-maze Object in place preference test	[81]
Astrocytic Gq-GPCR signaling activation with hM3DGq	Improved working memory	T-maze	[83]
Melanopsin-induced astrocytic optogenetic activation	Improved spatial memory	Place novelty preference test	[82]
Genetic deletion of S100β	Improved spatial memory	MWM test	[84]
Astrocytic A_2A_ receptor knockdown	Improved spatial memory	MWM test	[88]
Astrocytic Gq-GPCR inhibition with iβARK	Impaired spatial working memory	Y-maze Novel object placement	[85]
Genetic deletion of IP_3_R2	Impaired spatial memory	Y-maze	[86]
Genetic deletion of astrocytic GABA_B_ receptor	Impaired working memory	T-maze Object in place preference test	[81]
Astrocytic A_2A_ receptor deletion	Impaired spatial memory	Y-maze Baited eight-radial arm maze	[87]
Inhibition of glycogenolysis	Impaired working memory	Spontaneous alternation in elevated plus-maze	[93]

MWM: Morris water maze.

By integrating findings from transgenic mice, optogenetic and chemogenetic, it can be argued that astrocytic Ca^2+^ signals as well as energy metabolism play critical roles in the formation of both working and spatial memory.

### 4.2. Recognition Memory

A reduction in endogenous mitochondrial ROS (reactive oxygen species) production within astrocytes in vivo has been demonstrated to result in compromised recognition memory [94], underscoring the essential role of brain energy metabolism in this type of memory. Moreover, the specific removal of the carrier receptor p75^NTR^, which is accountable for proBDNF uptake in astrocytes, interferes with memory functions such as recent object recognition and hinders late-phase LTP (L-LTP) [95]. 

Regarding the changes in brain energy metabolism, the existing literature has shown that the inhibition of hippocampal glycogenolysis hinders recognition memory, a deficit that is restored by the administration of exogenous L-lactate [96].

**Table 2 ijms-25-01899-t002:** Astrocyte involvement in recognition memory.

Proposed Astrocytic Mechanism	Outcome of the Astrocytic Manipulation	Test	Reference
Decreased mitochondrial ROS in astrocytes	Impaired recognition memory	NOR task	[94]
Astrocyte deletion of proBDNF uptake carrier receptor (p75NTR)	Impaired recognition memory	NOR task	[95]
Inhibition of glycogenolysis	Impaired recognition memory	NOR task	[96]

NOR: novel object recognition.

### 4.3. Contextual Memory

Through the observation of astrocytic Ca^2+^ activity on both single-cell and small cluster levels in awake mice, a recent study utilized in vivo two-photon Ca^2+^ imaging and optic fiber-based recordings to unveil the crucial role of changes in astrocytic activity in contextual learning [97]. This investigation discovered Ca^2+^ alterations in a cluster of astrocytes during a fear conditioning task [97]. The researchers discovered that this responsiveness of astrocytes depends on the activation of α7-nicotinic acetylcholine receptors (α7-nAChRs) by signals originating from cholinergic neurons [97], and the deletion of this receptor in astrocytes significantly impairs persistent fear memory [97].

Furthermore, activating Gq signaling in hippocampal astrocytes stimulates them, enhancing recent fear memory and inducing LTP through the release of D-serine [83]. ChR2 optogenetic astrocytic activation, which leads to increased intracellular Ca^2+^ by an influx from the extracellular space (in contrast to the Gq-GPCR activation-induced release of Ca^2+^ from internal stores), results in A_1_R activation via adenosine release, impairing recent fear memory [98]. Additionally, astrocytic Gq activation in the amygdala impairs fear memory, mediated by astrocyte-released adenosine-induced A_1_R-inhibition [99].

Apart from Gq signaling, the specific activation of Gi-GPCRs in astrocytes amplifies astrocyte-specific Ca^2+^ modifications in vivo [100,101]. Consistent with this idea, activating CA1 astrocytes through Gi-GPCR or Gi-coupled µ-opioid receptors boosts the recall of recent contextual memory associated with conditioned place preference (CPP) [102]. Gi-like astrocytic activation prompts the release of glutamate via K2P channels, increases the probability of glutamate release by stimulating presynaptic mGluR1s, and enhances early-phase LTP. This mechanism could potentially elucidate the process of acquiring memory linked to CPP [102].

In contrast to studies suggesting memory improvement with Gi activation, another study using the same Gi-like DREADD manipulation in hippocampal CA1 found impaired long-term but not short-term fear memory [103]. Likewise, blocking the astrocytic IP_3_R2 pathway selectively impedes remote fear memory and spatial memory, impacting L-LTP through the modulation of astrocyte-derived BDNF [104]. Another study employing IP_3_R2 ablation demonstrated impairments in long-term fear, spatial memory, and recognition memory [86].

Apart from calcium modulation, the exploration of the specific gliotransmitter released from astrocytes and the way in which astrocytes impact on neurons during the learning and memory process is captivating. Notably, hippocampal astrocytes receive cholinergic inputs via m1-AChRs regulating the release of BDNF, a crucial factor for neurogenesis and fear memory. The deletion of the m1-AChRs in the astrocytes of the hippocampus results in impaired fear memory and the maturation of newly formed neurons [76]. Furthermore, the knockout of astrocytic BDNF in adult individuals disrupts long-term fear memory, while leaving short-term fear memory unaffected. This impairment is restored by augmenting hippocampal astrocytic BDNF expression before the fear training phase [104].

Regarding metabolism, a decrease in the generation of L-lactate from astrocytes in the hippocampus hinders LTP and disrupts remote memory in an inhibitory avoidance task. This impairment can be restored by the administration of exogenous L-lactate [61]. The same disruptions in memory function have been observed with the downregulation of monocarboxylate transporters (MCTs), responsible for the lactate transfer from astrocytes to neurons [61]. Another study demonstrated that pyruvate and the ketone bodies can effectively substitute for lactate, restoring the memory deficits induced by glycogenolysis inhibition or the downregulation of astrocytic MCTs [105].

Furthermore, stress triggers glucocorticoid release that governs energy homeostasis and contributes to emotional memory [106]. In line with this idea, knockout mice for astrocytic glucocorticoid receptors exhibit compromised fear memory and impaired brain glucose metabolism [107].

**Table 3 ijms-25-01899-t003:** Astrocyte involvement in contextual memory.

Proposed Astrocytic Mechanism	Outcome of the Astrocytic Manipulation	Test	Reference
Activation of astrocytic Gq-GPCR signalling with hM3DGq or opto-α-1AR in CA1	Improved fear memory	Fear conditioning	[83]
Gi-coupled µ-opioid receptor activation in astrocytes	Improved contextual memory	CPP test	[102]
Astrocytic α7-nAChRs deletion	Impaired fear memory	Fear conditioning	[97]
Elimination of astrocytic m1-AChRs	Impaired fear memory	Fear conditioning	[76]
Optogenetic activation of astrocytes	Impaired fear memory	Fear conditioning	[98]
Astrocytic Gq-GPCR signalling activation with hM3DGq in the amygdala	Impaired fear memory	Fear conditioning	[99]
Activation of astrocytic Gi-GPCR signalling with hM4DGi	Impaired fear memory	Fear conditioning	[103]
Astrocytic genetic deletion of IP_3_R2	Impaired fear memory (and spatial memory)	Fear conditioning (and MWM test)	[104]
Astrocytic genetic deletion of IP_3_R2	Impaired fear memory	Fear conditioning	[86]
Decrease in the generation of L-lactate	Impaired contextual memory	Inhibitory avoidance task	[61]
MTC4 or MCT1 deletion in the hippocampus	Impaired contextual memory	Inhibitory avoidance task	[105]
Astrocytic glucocorticoid receptor deletion	Impaired fear memory	Fear conditioning	[107]

CPP: conditioned place preference; MWM: Morris water maze.

## 5. Astrocytic Calcium Signalling as a Key Element in AD Pathology

Most neurodegenerative diseases, particularly AD, profoundly affect learning and memory. AD alone constitutes approximately 60–70% of all dementia cases, making it the most prevalent type of dementia. According to the latest report from the Alzheimer’s Association, deaths related to AD increased by 145% between 2000 and 2019. Given that AD is associated with aging and global life expectancy is on the rise, these numbers are projected to increase. This high prevalence establishes AD as a significant contemporary global health concern.

The molecular and cellular mechanisms underlying the development of AD and the initial events preceding cognitive decline remain poorly understood. Traditionally, research has primarily focused on the primary features of the disease: the accumulation of amyloid-β (Aβ) and the formation of neurofibrillary tangles, and their detrimental effects on neuronal function. However, amidst the search for a cure, an essential yet often overlooked aspect is the critical role of the dynamic interactions between neurons and astrocytes in brain function. Indeed, astrocytes experience changes in AD, where disruptions in astrocytic Ca^2+^ activity, glutamate uptake, and mitochondrial respiration are observed, potentially leading to neural damage [108,109]. Given the significance of gliotransmission in brain physiology and the well-documented disturbances in astrocytic Ca^2+^ homeostasis in AD mice [110,111], targeted investigations are warranted to uncover novel therapeutic targets and early AD indicators associated with astrocytic Ca^2+^ signalling.

According to studies conducted in mouse models of AD based on mutations in APP and PS1 genes, both excessive and reduced levels of Ca^2+^ activity have been observed at the neuronal level, with excessive activity predominating in the early stages of the disease, attributed to soluble Aβ [112,113,114,115]. In AD mice, the presence of amyloidosis has generally been linked with increased astrocytic Ca^2+^ activity [111,116]. In certain mouse models, such as APPPS1 mice, this elevated activity involves abnormal purinergic Ca^2+^ signalling [117,118], although recent research has also indicated a decrease in sensory-evoked astrocyte responsiveness in AD mice with APP and PS1 mutations [119]. Correspondingly, a diminished astrocytic Ca^2+^ response to movement has been observed in the neocortex of awake-behaving 15-month-old tg-ArcSwe mice [120], and a causal relationship has been established in APP^NL-F^ mice between network hyperactivity and impaired Ca^2+^ signalling in astrocytes in early disease stages [121].

However, it remains unclear how Ca^2+^ activity evolves in astrocytes as the disease progresses and its correlation with alterations in memory formation and retrieval. This issue has been recently tackled in a very elegant study, revealing that Ca^2+^ signalling in astrocytes from APS2APP AD transgenic mice undergoes significant changes throughout the course of AD progression [122]. More precisely, there is a transition from a trend toward elevated spontaneous Ca^2+^ activity at 3 months of age to significant Ca^2+^ hypoactivity at 6 months of age, aligning with the emergence of plaque deposition and gliosis. This impaired Ca^2+^ signalling corresponds with diminished astrocyte-mediated LTP and tactile memory in AD mice at 6 months, eventually resulting in memory decline by the time they reach 8 months of age.

The above-mentioned studies highlight the central role of astrocytes in AD pathology, pointing to astrocytic Ca^2+^ modulation as a potential primary target for the development of novel AD therapies.

## 6. Conclusions

Based on the evidence presented, it can be asserted that astrocytes play a pivotal role in learning and memory processes. Specifically, astrocytes contribute by providing metabolic support to neurons, helping in energy-demanding sequential processes such as gene transcription, protein synthesis, as well as post-translational modifications. Additionally, the regulation of astrocytic intracellular Ca^2+^ emerges as a crucial element in memory modulation. The concentration of astrocyte intracellular Ca^2+^ is controlled through Gq-coupled (e.g., CB_1_Rs, GABA_B_Rs, and IP_3_R2s), Gi-coupled (e.g., µ-opioid receptors), or Gs-coupled signaling (e.g., A_2_Ars), and these signalling pathways play a critical role in both short-term and long-term memory. In this sense, future studies employing astrocytic Ca^2+^ imaging, along with genetically encoded fluorescent markers for gliotransmitters, alongside neuronal recordings coupled with astrocyte manipulation in memory-related brain areas of awake rodents across various stages of learning and memory, are positioned to offer new perspectives on these unresolved questions.

In summary, astrocytes exert various influences on learning and memory. The precise dynamics of astrocytic Ca^2+^ are increasingly recognized as central contributors to memory formation and retrieval. Moreover, astrocytes regulate brain rhythms crucial for cognition, and the cooperative metabolic interactions between astrocytes and neurons are imperative for memory consolidation.

## Figures and Tables

**Figure 1 ijms-25-01899-f001:**
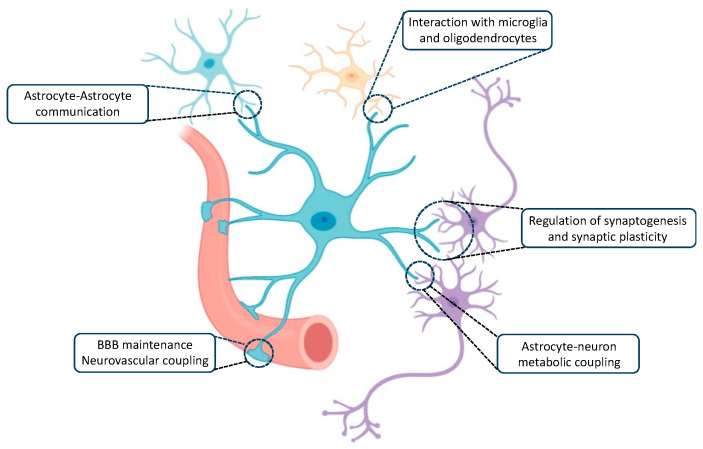
Schematic representation of various supportive and neuroprotective roles of astrocytes under physiological conditions. BBB: Blood brain barrier.

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
