# Peer review of "Essential Role of Astrocytes in Learning and Memory"

_ijms, 2024, doi:10.3390/ijms25031899_

Round 1
Reviewer 1 Report
Comments and Suggestions for Authors
The manuscript provides a comprehensive and insightful exploration of the crucial role played by astrocytes in the regulation of synaptic transmission, particularly in the context of learning and memory processes within the central nervous system (CNS). The authors effectively highlight the significance of astrocytes as integrators of learning- and memory-relevant information, shedding light on their involvement in synaptic connection strength and the cellular basis of memory formation and retrieval.
One commendable aspect of the review is the clear delineation of the inputs and outputs of astrocytes, contributing to a thorough understanding of their functional mechanisms. The authors adeptly discuss the impact of manipulating astrocytes on behavior across diverse learning paradigms, providing a nuanced perspective on the multifaceted influence of astrocytes on learning and memory.
The focus on astrocytic calcium dynamics as central contributors to memory formation and retrieval is particularly noteworthy. The manuscript effectively synthesizes current knowledge on astrocytes' role in regulating brain rhythms essential for cognition, emphasizing the intricate astrocyte-neuron metabolic cooperation required for memory consolidation.
-
In-Depth Exploration:
- Expand on specific examples or case studies that illustrate the role of astrocytes in learning and memory processes. This could provide readers with concrete instances and enhance the practical implications of the discussed concepts.
-
Integration of Recent Studies:
- Incorporate the latest studies and findings in the field to ensure the review reflects the most up-to-date understanding of astrocyte functions in synaptic transmission and memory processes.
-
Connections to Clinical Relevance:
- Explore potential connections between the insights provided in the review and clinical applications or implications. Discussing how this knowledge may impact neurodegenerative diseases or cognitive disorders could add a practical dimension to the review.
-
Final Proofreading:
- Conduct a final proofreading to ensure grammatical accuracy, consistency in referencing, and adherence to formatting guidelines.
By incorporating these suggestions, the manuscript can be further refined to enhance its impact and contribute even more effectively to the understanding of astrocytes in learning and memory processes.
Overall, this review significantly contributes to the existing literature by providing a well-organized and in-depth analysis of the role of astrocytes in learning and memory processes. The integration of various aspects, from gliotransmitter release to astrocyte-neuron metabolic coupling, enhances the overall quality of the manuscript and makes it a valuable resource for researchers and scholars interested in neurobiology and cognitive science.
Author Response
We thank the reviewer its helpful comments to increase the quality of the paper.
The manuscript provides a comprehensive and insightful exploration of the crucial role played by astrocytes in the regulation of synaptic transmission, particularly in the context of learning and memory processes within the central nervous system (CNS). The authors effectively highlight the significance of astrocytes as integrators of learning- and memory-relevant information, shedding light on their involvement in synaptic connection strength and the cellular basis of memory formation and retrieval.
One commendable aspect of the review is the clear delineation of the inputs and outputs of astrocytes, contributing to a thorough understanding of their functional mechanisms. The authors adeptly discuss the impact of manipulating astrocytes on behavior across diverse learning paradigms, providing a nuanced perspective on the multifaceted influence of astrocytes on learning and memory.
The focus on astrocytic calcium dynamics as central contributors to memory formation and retrieval is particularly noteworthy. The manuscript effectively synthesizes current knowledge on astrocytes' role in regulating brain rhythms essential for cognition, emphasizing the intricate astrocyte-neuron metabolic cooperation required for memory consolidation.
We sincerely appreciate the positive and encouraging words of the reviewer.
In-Depth Exploration:
Expand on specific examples or case studies that illustrate the role of astrocytes in learning and memory processes. This could provide readers with concrete instances and enhance the practical implications of the discussed concepts.
Integration of Recent Studies:
Incorporate the latest studies and findings in the field to ensure the review reflects the most up-to-date understanding of astrocyte functions in synaptic transmission and memory processes.
Connections to Clinical Relevance:
Explore potential connections between the insights provided in the review and clinical applications or implications. Discussing how this knowledge may impact neurodegenerative diseases or cognitive disorders could add a practical dimension to the review.
We thank the reviewer for offering us all those important points. In order to cover all of them together a new section regarding “Astrocytic calcium signalling as a key element in Alzheimer’s disease pathology” has been added (please see section 5, lines 410-454). In that new section we have covered specific examples of the important involvement of astrocytes in the cognitive deficiency observed in Alzheimer’s disease, supported by recent studies (please see references 113-116) and we consider that this new section offers to the review the clinical point of view suggested by the reviewer.
Final Proofreading:
Conduct a final proofreading to ensure grammatical accuracy, consistency in referencing, and adherence to formatting guidelines.
A deep proofreading has been conducted.
Reviewer 2 Report
Comments and Suggestions for Authors
In this paper these authors review the role of astrocytes in learning and memory. First, their role in supporting the structure and function of synapses and blood capillaries and maintaining the blood-brain barrier is presented and illustrated in a helpful figure. Astrocytes can communicate with other astrocytes via a network of gap junctions while regulating pre- and post-synaptic transmission - the molecular mechanisms used to do this are detailed. As well as regulating synaptic plasticity, astrocytes also influence levels and effectiveness of neurogenesis in the dentate nuclei and olfactory bulbs. The cellular basis of learning and memory lies in the strength and plasticity of synaptic connections. Astrocytes influence this by releasing gliotransmitters such as glutamate, ATP, and D-serine, triggered by rises in astrocytic calcium activity and astrocyte-neuron metabolic coupling via lactate release and uptake. Astrocytes act to influence glutamate GABA, and monoamine modultory activity so integrating learning- and memory-relevant information. Finally, the animal literature demonstrating a vital role of astrocytes in working and spatial memory, recognition, and contextual memory is presented in the text and as tables and discussed. The influence of manipulating astrocytes on behaviour during learning paradigms is detailed. It is concluded that astrocytes play an essential role in influence learning and memory although they are not excitable cells and act on a slower time scale than neurones.
This is a well written, convincing, and interesting review and I found it highly educational. My only criticism is that non-molecular biologists may struggle at times as, for example, there is no discussion of the roles of the various G-coupled pathways by which astrocytes are stated to influence synaptic function. The review is likely to be widely read and cited.
Comments on the Quality of English LanguageThe text would benefit from proof reading by a native English speaker and / or a grammar check as there are a few grammatical errors.
Author Response
We thank the reviewer its helpful comments to increase the quality of the paper.
In this paper these authors review the role of astrocytes in learning and memory. First, their role in supporting the structure and function of synapses and blood capillaries and maintaining the blood-brain barrier is presented and illustrated in a helpful figure. Astrocytes can communicate with other astrocytes via a network of gap junctions while regulating pre- and post-synaptic transmission - the molecular mechanisms used to do this are detailed. As well as regulating synaptic plasticity, astrocytes also influence levels and effectiveness of neurogenesis in the dentate nuclei and olfactory bulbs. The cellular basis of learning and memory lies in the strength and plasticity of synaptic connections. Astrocytes influence this by releasing gliotransmitters such as glutamate, ATP, and D-serine, triggered by rises in astrocytic calcium activity and astrocyte-neuron metabolic coupling via lactate release and uptake. Astrocytes act to influence glutamate GABA, and monoamine modultory activity so integrating learning- and memory-relevant information. Finally, the animal literature demonstrating a vital role of astrocytes in working and spatial memory, recognition, and contextual memory is presented in the text and as tables and discussed. The influence of manipulating astrocytes on behaviour during learning paradigms is detailed. It is concluded that astrocytes play an essential role in influence learning and memory although they are not excitable cells and act on a slower time scale than neurones.
We sincerely appreciate the positive words of the reviewer.
This is a well written, convincing, and interesting review and I found it highly educational. My only criticism is that non-molecular biologists may struggle at times as, for example, there is no discussion of the roles of the various G-coupled pathways by which astrocytes are stated to influence synaptic function. The review is likely to be widely read and cited.
We thank the reviewer to point this issue into our attention. A new sentence has been added in order to explain the intracellular pathway activated upon Gq-GPCR stimulation that leads to calcium elevation (please see lines 287-293).
The text would benefit from proof reading by a native English speaker and / or a grammar check as there are a few grammatical errors.
All the text has been checked.